# Object DGCNN:
# 3D Object Detection using Dynamic Graphs

**Yue Wang**
Massachusetts Institute of Technology
`yuewang@csail.mit.edu`

**Justin Solomon**
Massachusetts Institute of Technology
`jsolomon@mit.edu`

## Abstract

3D object detection often involves complicated training and testing pipelines, which require substantial domain knowledge about individual datasets. Inspired by recent non-maximum suppression-free 2D object detection models, we propose a 3D object detection architecture on point clouds. Our method models 3D object detection as message passing on a dynamic graph, generalizing the DGCNN framework to predict a set of objects. In our construction, we remove the necessity of post-processing via object confidence aggregation or non-maximum suppression. To facilitate object detection from sparse point clouds, we also propose a set-to-set distillation approach customized to 3D detection. This approach aligns the outputs of the teacher model and the student model in a permutation-invariant fashion, significantly simplifying knowledge distillation for the 3D detection task. Our method achieves state-of-the-art performance on autonomous driving benchmarks. We also provide abundant analysis of the detection model and distillation framework.

Methods for 3D object detection have progressed rapidly, yielding deployable autonomous driving perception systems. Following common practice in 2D vision, 3D object detection often employs complex training and testing pipelines including many post-processing operations to achieve superior performance. These operations are typically non-parallelizable and inefficient even with modern deep learning frameworks, implying a steep trade-off between between efficiency and effectiveness.

Modern methods usually employ two stages [1, 2], including a region proposal network [3] that can introduce significant training overhead. Subsequent efforts simplify this pipeline for 3D object detection. PointPillars [4] introduces a one-stage anchor-based design, simplfying training. PillarOD [5] and CenterPoint [6] improve the one-stage model by making per-pillar predictions, that is, one prediction per point on the ground plane. They assign ground-truth bounding boxes to multiple outputs while training to ease optimization. However, they predict redundant boxes, which can overlap in the same positions; extra boxes are eliminated *a posteriori* using non-maximum suppression (NMS). It remains elusive to remove hand-designed components like NMS in training and testing.

We introduce Object DGCNN, a streamlined architecture for 3D object detection from point clouds. Like DETR for 2D object detection [7], we predict a set of bounding boxes from the raw data, enabling an NMS-free pipeline that achieves real-time performance. A critical new component is to treat each object query as a point in a set whose embedding is learned using DGCNN [8]. Compared to the self-attention module [9] in DETR, DGCNN leverages a *sparse* set of object relations, which reflects the real object distribution in the scene. In contrast to PointPillars [4], PillarOD [5], and CenterPoint [6], our method *does not* require post-processing.

We also provide a knowledge distillation approach customized to 3D object detection. Existing methods typically distill dense feature maps from a teacher model to a student model, whose training objective does not necessarily capture 3D object detection performance [10]. In contrast, we propose set-to-set distillation training that aligns the outputs of the teacher and the student in a permutation-invariant fashion. This process is enabled by the unified Object DGCNN architecture. In addition to

35th Conference on Neural Information Processing Systems (NeurIPS 2021).

obtaining better performance, through this process our model can benefit from privileged information (e.g., dense point clouds) only available at training time.

**Contributions.** We summarize our key contributions as follows:

- We propose a post-processing-free 3D object detection model achieving state-of-the-art performance. To our knowledge, this is the first NMS-free 3D object detector.
- We generalize DGCNN to model objects as a point set. The DGCNN module outperforms its self-attention counterpart thanks to its sparse structure.
- We propose a set-to-set distillation method for 3D object detection. In our construction, knowledge distillation on object detection simply penalizes differences between the outputs of the teacher model and the student model.
- We show our model can use privileged information (such as dense point clouds) that is naturally available at training time to improve the model performance at inference time.
- We release our code to promote reproducibility and future research. [1]

# 1   Related Work

**2D object detection.** Object recognition research has been transitioning from models with hand-crafted components to models with limited post-processing. One-stage detectors [11–13] remove the complicated region proposal networks in two-stage objectors [3, 14], yielding more efficient training and testing. Anchor-free methods [15, 16] further simplify the one-stage pipeline by shifting from per-anchor prediction to per-pixel prediction. However, these methods still make dense predictions and rely on NMS to reduce redundancy. To alleviate this issue, DETR [7] formulates object detection as a set-to-set prediction problem. It introduces a set-to-set loss that implicitly penalizes redundant boxes, removing the necessity of post-processing. To accelerate convergence, Deformable DETR [17] proposes deformable self-attention and streamlines the optimization process. Our method also formulates 3D object detection as set prediction, but with a customized design for 3D.

**3D object detection.** VoxelNet [18] generalizes one-stage object detection to 3D. It uses 3D dense convolutions to learn representations on voxelized point clouds, which is too inefficient to capture fine-grained features. To address that, PIXOR [19] and PointPillars [4] project points to a birds-eye view (BEV) and operate on 2D feature maps; PointNet [20] aggregates features within each BEV pixel. We use a variant of PointPillars [4] for 3D detection (§3). These methods are efficient but drop information along the vertical axis. To accompany the BEV projection, MVF [21] adds a spherical projection. PillarOd [5] and CenterPoint [6] use pillar-centric object detection, making predictions per BEV pixel (pillar) rather than per anchor. These anchor-free methods simplify 3D object detection while maintaining efficiency. Beyond SSD-style [11] one-stage models, Complex-YOLO [22] extends YOLO to 3D for real-time perception. PointRCNN [23] employs a two-stage architecture for high-quality detection. To improve representations of two-stage models, PVRCNN [2] proposes a point-voxel feature set abstraction layer to leverage the flexible receptive fields of PointNet-based networks. Unlike works on point clouds, LaserNet [24] operates on raw range scans with comparable performance. [25–27] combine point clouds with camera images. Frustum-PointNet [28] leverages 2D object detectors to form a frustum crop of points and then uses PointNet to aggregate features. [29] describes an end-to-end learnable architecture that exploits continuous convolutions to fuse feature maps. VoteNet [30, 31] generalizes Hough voting [32] for 3D object detection in point clouds. DOPS [33] extends VoteNet and predicts 3D object shapes. In addition to visual input, [34] shows that high-definition (HD) maps can boost performance of 3D object detectors. [35] argues that multi-tasking can learn better representations than single-tasking. Beyond supervised learning, [36] learns a perception model for unknown classes.

**DGCNN.** DGCNN [8] pioneered learning point cloud representations via dynamic graphs. It models point clouds as connected graphs, which are dynamically built using $k$-nearest neighbors in the latent space. DGCNN learns per-point features through message passing. However, it operates on point clouds for single object recognition and semantic segmentation. One of our key contributions is to generalize DGCNN to model scene-level object relations for 3D detection.

**Knowledge distillation (KD).** KD compresses knowledge from an ensemble of models into a single smaller model [37]. [38] generalizes this idea and combines it with deep learning. KD transfers

---

[1] `https://github.com/WangYueFt/detr3d`

knowledge from a teacher model to a student model by minimizing a loss, in which the target is the distribution of class probabilities induced by the teacher. [39–50] improve knowledge distillation for classification. Beyond image classification, KD has been extended to improve object detection. [51] leverages FitNets for object detection, addressing obstacles such as class imbalance, loss instability, and feature distribution mismatch. [52] distills between region proposals, accelerating training with added instability. To address this issue, [53] uses fine-grained representation imitation using object masks. [54] uses KD to tackle a continual learning problem.

**Privileged information.** [55] introduces the framework of learning with privileged information in the context of support vector machines (SVMs), wherein additional information is accessible during training but not testing. [56] unifies KD and learning using privileged information theoretically. [57] identifies practical applications, e.g., transferring knowledge from localized data to non-localized data, from high resolution to low resolution, from color images to edge images, and from regular images to distorted images. To mediate uncertainty and improve training efficiency, [58] makes the variance of Dropout [59] a function of privileged information. We extend these methods to 3D data, in which privileged information consists of dense point clouds aggregated from LiDAR sequences.

## 2 Overview

Our target application of object detection differs from the recognition and segmentation tasks considered for DGCNN. Our point clouds typically contain too many points to apply DGCNN and its peers directly to the entire scene. Moreover, the size of our output set, a small set of bounding boxes, differs from the size of our input set, a huge set of points in $\mathbb{R}^3$.

Following state-of-the-art in large-scale object detection, our pipeline learns a grid-based intermediate representation to capture local features (§3). We test two standard learning-based methods for collecting local point cloud features on a birds-eye view (BEV) grid. While in principle it might be possible to avoid grids entirely in our pipeline, this BEV representation is far more efficient and—as observed in previous work—is sufficient to find objects reliably in autonomous driving, where there is likely only one object above any given grid cell on the ground plane.

Our main architecture contribution is the Object DGCNN pipeline (§4), which transitions from this BEV grid of features to a *set* of object bounding boxes. Object DGCNN draws inspiration from the DGCNN architecture; its layers alternate between local feature transformations and $k$-nearest neighbor aggregation to capture relationships between objects. Unlike conventional DGCNN, however, Object DGCNN incorporates features from the BEV grid in each of its layers; each layer incorporates several queries into the BEV to refine object position estimates. The output of Object DGCNN is a *set* of objects in the scene. We use a permutation-invariant loss (10) to measure divergence from the ground truth set of objects.

The pipeline above does not require hand-designed post-processing like NMS; our output boxes are usable directly for object detection. Beyond simplifying the object detection pipeline, this allows us to propose object detection-specific distillation procedures (§6.3) that further improve performance. These use one network to train another, e.g., to train a network operating on sparse point clouds to output features that imitate those learned by a network trained on denser, more detailed point clouds.

## 3 Local Features

We begin with a point cloud $\mathcal{X} = \{\boldsymbol{x}_1, \ldots, \boldsymbol{x}_i, \ldots, \boldsymbol{x}_N\} \subset \mathbb{R}^3$ with per-point features $\mathcal{F} = \{\boldsymbol{f}_1, \ldots, \boldsymbol{f}_i, \ldots, \boldsymbol{f}_N\} \subset \mathbb{R}^K$, ground-truth bounding boxes $\mathcal{B} = \{\boldsymbol{b}_1, \ldots, \boldsymbol{b}_j, \ldots, \boldsymbol{b}_M\} \subset \mathbb{R}^9$, and categorical labels $\mathcal{C} = \{c_j, \ldots, c_j, \ldots, c_M\} \subset \mathbb{Z}$. Each $\boldsymbol{b}_j$ contains position, size, heading angle, and velocity in the birds-eye view (BEV); our architecture aims to predict these boxes and their labels from the point cloud and its features.

As an initial step, modern 3D object detection models scatter points into either BEV pillars or 3D voxels and then use convolutional neural networks to extract features on a grid. This strategy accelerates object detection for large point clouds. We test two neural network architectures for BEV feature extraction, detailed below.

PointPillars [4] maps sparse point clouds onto a dense BEV pillar map on which 2D convolutions can be applied. Suppose $F_P(i)$ returns the points in pillar $i$, that is, the set of points in a vertical column

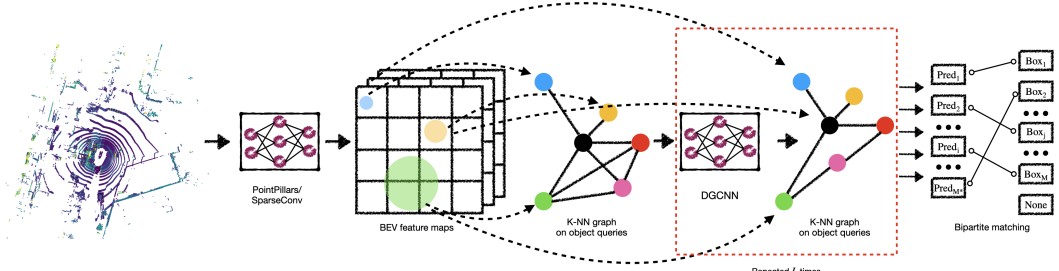

Figure 1: Overview. Point cloud features are learned in BEV, followed by $L$ DGCNNs to model object relations. We predict a set of bounding boxes and compute loss in a one-to-one manner.

above point $i$ on the ground. When collecting features from points to pillars, multiple points can fall into the same pillar. In this case, PointNet [20] (PN) is used to obtain pillar-wise features:

$$\boldsymbol{f}_i^{\text{pillar}} = \text{PN}(\{\boldsymbol{f}_j | \boldsymbol{x}_j \in F_P(\boldsymbol{p}_i)\}), \tag{1}$$

where $\boldsymbol{f}_i^{\text{pillar}}$ is the feature of pillar $\boldsymbol{p}_i$. We set the features for empty pillars to $\boldsymbol{0}$. This results in a dense 2D grid $\mathcal{F}^{\text{pillar}} \subset \mathbb{R}^{H^{\text{p}} \times W^{\text{p}} \times C^{\text{p}}}$, where $H^{\text{p}}, W^{\text{p}}$ and $C^{\text{p}}$ are the height, width, number of channels of this 2D pillar map, respectively. Multiple stacked convolutional layers further embed the pillar features to the final feature map $\mathcal{F}^{\text{d}} \subset \mathbb{R}^{H^{\text{d}} \times W^{\text{d}} \times C^{\text{d}}}$.

An alternative BEV embedding is SparseConv [60]. If $F_V(i)$ returns the set of points in voxel $i$, SparseConv collects point-wise features into voxel-wise features by

$$\boldsymbol{f}_i^{\text{voxel}} = \text{PN}(\{\boldsymbol{f}_j | \boldsymbol{x}_j \in F_V(i)\}), \tag{2}$$

where $\boldsymbol{f}_i^{\text{voxel}}$ contains the features of voxel $i$. In contrast to PointPillars, SparseConv conducts 3D sparse convolutions to refine the voxel-wise features. Finally, we compress these sparse voxels to a BEV 2D grid by filling empty voxels with zeros and averaging along the $z$-axis. For ease of notation, we also denote the resulting 2D grid $\mathcal{F}^{\text{d}} \subset \mathbb{R}^{H^{\text{d}} \times W^{\text{d}} \times C^{\text{d}}}$.

## 4    Object DGCNN

After obtaining the BEV features $\mathcal{F}^{\text{d}}$ using one of the architectures above, we predict a set of bounding boxes as well as a label for each box. The key difference between our architecture and most recent 3D object detection methods is that ours produces a *set* of bounding boxes rather than a box per grid cell followed by NMS, as in [5, 6]. Hence, we need to transition from a grid of per-pillar features to an unordered set of objects; we detail our approach below. We address two key issues: prediction of the bounding boxes and evaluation of the loss.

**Desiderata.** Object DGCNN uses a DGCNN-inspired architecture but incorporates grid-based BEV features, built on the philosophy that local features (§3) are reasonable to store on a dense grid, but object predictions are better modeled using sets. Hence, we require a new architecture and set-to-set loss that encourage bounding box diversity.

Object DGCNN uses $L$ layers that follow a series of set-based computations to produce bounding box predictions from the BEV feature maps. Each layer employs the following steps (Figure 1):

1. predict a set of query points and attention weights;
2. collect BEV features from keypoints determined by the queries; and
3. model object-object interactions via DGCNN.

Each layer results in a more refined set of bounding box predictions, one per query. At the end of these layers, we match the prediction set with the ground-truth set in a one-to-one fashion and evaluate a set-to-set object detection loss.

**Single layer.** Inspired by DETR [7], each layer $\ell \in \{0, \ldots, L-1\}$ of Object DGCNN operates on a set of *object queries* $\mathcal{Q}_\ell = \{\boldsymbol{q}_{\ell 1}, \ldots, \boldsymbol{q}_{\ell M^*}\} \subset \mathbb{R}^Q$, producing a new set $\mathcal{Q}_{\ell+1}$. Although queries are fully learnable, our intuition is that they represent progressively refined object positions.

The initial set of object queries $\mathcal{Q}_0$ is learned jointly with the neural network weights, yielding a dataset-specific prior. Beyond this fixed initial set, below we detail how to incorporate scene information to obtain $\mathcal{Q}_{\ell+1}$ from $\mathcal{Q}_\ell$ using an approach inspired by DGCNN [8] and deformable self-attention [17]. For notational convenience, we drop the $\ell$ subscript.

Starting from each query $\boldsymbol{q}_i$ (or, without the index dropped, $\boldsymbol{q}_{\ell i}$), we decode a reference point $\boldsymbol{p}_i \in \mathbb{R}^2$, a set of offsets $\{\boldsymbol{\delta}_{i0}, \ldots, \boldsymbol{\delta}_{iK}\} \subset \mathbb{R}^2$, and a set of attention weights $\{w_{i0}, \ldots, w_{iK}\} \subset \mathbb{R}$:

$$\boldsymbol{p}_i = \Phi_{\mathrm{ref}}(\boldsymbol{q}_i), \qquad \{\boldsymbol{\delta}_i^0, \ldots, \boldsymbol{\delta}_i^k, \ldots \boldsymbol{\delta}_i^K\} = \Phi_{\mathrm{neighbor}}(\boldsymbol{q}_i),$$
$$\{w_i^0, \ldots, w_i^k, \ldots w_i^K\} = \Phi_{\mathrm{atten}}(\boldsymbol{q}_i), \tag{3}$$

where $\Phi_{\mathrm{ref}}$, $\Phi_{\mathrm{neighbor}}$, and $\Phi_{\mathrm{atten}}$ are shared neural networks among the queries. We think of $\boldsymbol{p}_i$ as a hypothesis for the center of the $i$-th object; the $\boldsymbol{\delta}$'s represent the positions of $K$ informative points relative to the position of the object that determine its geometry.

Next, we collect a BEV feature $\boldsymbol{f}_{ik}$ associated to each neighbor point $\boldsymbol{p}_{ik} = \boldsymbol{p}_i + \boldsymbol{\delta}_{ik}$ :

$$\boldsymbol{f}_{ik} = f_{\mathrm{bilinear}}(\mathcal{F}^d, \boldsymbol{p}_i + \boldsymbol{\delta}_{ik}), \tag{4}$$

where $f_{\mathrm{bilinear}}$ bilinearly interpolates the BEV feature map $\mathcal{F}^d$. Note this step is the interaction between our set-based architecture manipulating query points $\boldsymbol{q}_i$ and the grid-based feature map $\mathcal{F}^d$. We then aggregate a single object query feature $\boldsymbol{f}_i^o$ from the $\boldsymbol{f}_{ik}$s:

$$\boldsymbol{f}_i^o = \sum_k \frac{e^{w_{ik}}}{\sum_k e^{w_{ik}}} \boldsymbol{f}_{ik}. \tag{5}$$

This generates scene-aware features; each object query "attends" to a certain area in the scene.

In the current layer $\ell$, the queries have not yet interacted with each other. To incorporate neighborhood information in object detection estimates, we use DGCNN-style operations to model a sparse set of relations. We construct a graph between the queries using a nearest neighbor search in feature space. In particular, we connect each query feature $\boldsymbol{f}_i^o$ to its 16 nearest neighbors as ablated in Table 7. Identically to DGCNN, we learn a feature per edge $e_{ij}$ and then aggregate back to the vertices $i$ to produce the new set of object queries. In detail, we write:

$$\boldsymbol{q}_{(\ell+1)i} = \max_{\mathrm{edges}\ e_{ij}} \Phi_{\mathrm{edge}}(\boldsymbol{f}_i^o, \boldsymbol{f}_j^o), \tag{6}$$

where $\max$ denotes a channel-wise maximum and $\Phi_{\mathrm{edge}}$ is a neural network for computing edge features. This completes our layer for computing $\mathcal{Q}_{\ell+1}$ from $\mathcal{Q}_\ell$. Optionally, we repeat this last step multiple times, in effect applying DGCNN to the features $\boldsymbol{f}_i^o$ to get the point set $\mathcal{Q}_{\ell+1}$.

**Set-to-set loss.** After $L$ Object DGCNN layers as described above, we are left with a set of $M^*$ queries $\mathcal{Q}_L$ used to predict our bounding boxes. For each query $\boldsymbol{q}_{Li}$, we use a classification network to predict a categorical label $\hat{c}_i$ and a regression network to predict bounding box parameters $\hat{\boldsymbol{b}}_i$. Our final task is to assign the predictions to the ground-truth boxes and compute a set-to-set loss.

Most object detection models minimize a loss $\mathcal{L}_{\mathrm{od}}$ given by

$$\mathcal{L}_{\mathrm{od}} = \sum_{j=1}^{\hat{M}} -\log \hat{p}_{\hat{\sigma}(j)}(\hat{c}_j) + \mathbb{1}_{\{c_{\hat{\sigma}(j)} \neq \varnothing\}} \mathcal{L}_{\mathrm{box}}(\hat{\boldsymbol{b}}_j, \boldsymbol{b}_{\hat{\sigma}(j)}), \tag{7}$$

where $\hat{M} = H^d * W^d$, $\hat{\sigma}(*)$ returns the corresponding index of the ground-truth bounding box, $\hat{p}_{\hat{\sigma}(j)}(c_j)$ is the probability of class $c_{\hat{\sigma}(j)}$ for the prediction with index $\sigma(j)$, $\varnothing$ denotes an invalid box, and $\mathcal{L}_{\mathrm{box}}$ is typically the $\mathcal{L}_1$ distance. Different matchings $\hat{\sigma}$ yield different optimization landscapes and hence different prediction models. Pillar-OD [5] and CenterPoint [6] employ a simple $\hat{\sigma}$ to determine the ground-truth box used to evaluate the box predicted at BEV pixel $j$:

$$\hat{\sigma}_{\mathrm{overlap}}(j) = \begin{cases} j', & \text{if } \boldsymbol{b}_{j'} \text{ overlaps with BEV pixel } j; \\ \varnothing, & \text{otherwise.} \end{cases} \tag{8}$$

This strategy can assign a box to multiple nearby BEV pixels. This one-to-many assignment provides dense supervision for the object detector and eases optimization. Since the training objective

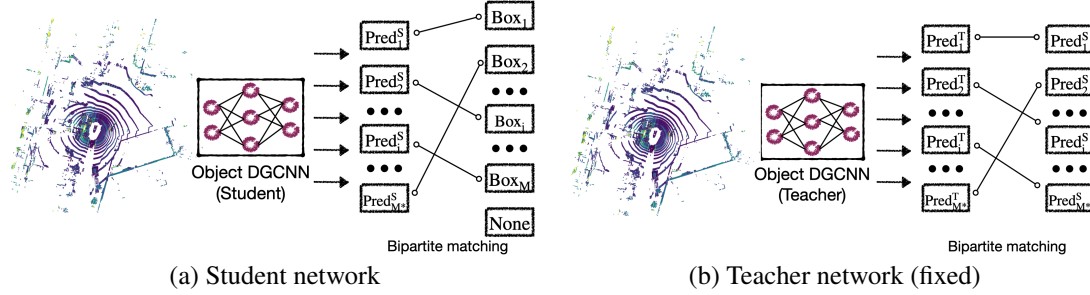

(a) Student network          (b) Teacher network (fixed)

Figure 2: The set-to-set distillation pipeline. The student network is trained with the ground-truth supervision as well as with the supervision from a fixed teacher network.

encourages each BEV pixel to predict the same surrounding box, however, redundant boxes are inevitable. So, NMS is usually required to remove redundant boxes at inference time.

Rather than performing dense predictions in the BEV, we make per-query predictions. Typically, $M^*$ is much larger than the number of ground-truth boxes $M$. To account for this difference, we pad the set of ground-truth boxes with $\varnothing$s (no object) up to $M^*$. Following [7], we use an objective built on an optimal matching between these two sets. We define the optimal bipartite matching as

$$\sigma^* = \arg\min_{\sigma \in \mathcal{P}} \sum_{j=1}^{M} -\mathbb{1}_{\{c_j \neq \varnothing\}} \hat{p}_{\sigma(j)}(c_j) + \mathbb{1}_{\{c_j = \varnothing\}} \mathcal{L}_{\text{box}}(\boldsymbol{b}_j, \hat{\boldsymbol{b}}_{\sigma(j)}), \tag{9}$$

where $\mathcal{P}$ denotes the set of permutations, $\hat{p}_{\sigma(j)}(c_j)$ is the probability of class $c_j$ for the prediction with index $\sigma(j)$, and $\mathcal{L}_{\text{box}}$ is the $\mathcal{L}_1$ loss for bounding box parameters. We use the Hungarian algorithm [61] to solve this assignment problem, as in [62, 7]. Our final set-to-set loss adapts (7):

$$\mathcal{L}_{\text{sup}} = \sum_{j=1}^{N} -\log \hat{p}_{\sigma^*(j)}(c_j) + \mathbb{1}_{\{c_j \neq \varnothing\}} \mathcal{L}_{\text{box}}(\boldsymbol{b}_j, \hat{\boldsymbol{b}}_{\sigma^*(j)}). \tag{10}$$

## 5 Distillation

Object DGCNN enables a new set-to-set knowledge distillation (KD) pipeline. KD usually involves a teacher model $\mathcal{T}$ and a student model $\mathcal{S}$. The common practice is to align the outputs of the student with those of the teacher using $\mathcal{L}_2$ distance or KL-divergence. In past 3D object detection methods, since final performance heavily relies on NMS and the predictions are post-processed to be a smaller set, distilling the teacher to the student is neither efficient nor effective. Since our set-based detection model is NMS-free, we can easily distill the information between models with homogeneous detection heads (per-query object detection head in our case). First, we train a teacher $\mathcal{T}$ using the method above with the loss in (10). Then, we train a student $\mathcal{S}$ with supervision given by $\mathcal{T}$ and the ground-truth. The class label and box parameters predicted by the teacher for each object query are $c_j^{\mathcal{T}}$ and $\boldsymbol{b}_j^{\mathcal{T}}$, respectively. The corresponding student outputs are $c_j^{\mathcal{S}}$ and $\boldsymbol{b}_j^{\mathcal{S}}$. We find an optimal matching between the output set of the teacher and that of the student:

$$\sigma_d^* = \arg\min_{\sigma_d \in \mathcal{P}} \sum_{j}^{N} -\log p_{\sigma_d(j)}(c_j^{\mathcal{T}}) + \mathcal{L}_{\text{box}}(\boldsymbol{b}_j^{\mathcal{T}}, \boldsymbol{b}_{\sigma_d(j)}^{\mathcal{S}}). \tag{11}$$

Then, the optimal matching's KD loss is given by

$$\mathcal{L}_{\text{distill}} = \sum_{j}^{N} -\log \hat{p}_{\sigma_d^*(j)}(c_j^{\mathcal{T}}) + \mathcal{L}_{\text{box}}(\boldsymbol{b}_j^{\mathcal{T}}, \boldsymbol{b}_{\sigma_d^*(j)}^{\mathcal{S}}). \tag{12}$$

So the overall loss during KD is $\mathcal{L} = \alpha \mathcal{L}_{\text{sup}} + \beta \mathcal{L}_{\text{distill}}$, where $\alpha$ and $\beta$ balance the supervised loss and distillation loss. In practice, we use $\alpha = \beta = 1$.

Table 1: Comparisons to recent works. Our method is robust to whether to use NMS. ∗: implementations with the same PointPillars backbone. ‡: implementations with the same SparseConv backbone.

| Method | NDS ↑ | mAP ↑ | mATE ↓ | mASE ↓ | mAOE ↓ | mAVE ↓ | mAAE ↓ | NMS |
|---|---|---|---|---|---|---|---|---|
| PointPillars [4] | 53.3 | 40.0 | - | - | - | - | - | ✓ |
| SSN [63] | 54.83 | 41.56 | - | - | - | - | - | ✓ |
| FreeAnchor [64] | 55.3 | 43.7 | - | - | - | - | - | ✓ |
| RegNetX-400MF-SECFPN [65] | 55.2 | 41.2 | - | - | - | - | - | ✓ |
| Pillar-OD [5] | 56.84 | 44.41 | - | - | - | - | - | ✓ |
| CenterPoint (pillar) [6] ∗ | 59.56 | 47.48 | **31.27** | **25.81** | 33.78 | 32.25 | 20.20 | ✓ |
| CenterPoint (pillar) [6] ∗ | 55.08 | 40.27 | 35.14 | 26.44 | 36.75 | 32.66 | 19.55 | |
| CenterPoint (voxel) [6] ‡ | **64.19** | **54.99** | 29.83 | 25.71 | 32.56 | 26.08 | **18.89** | ✓ |
| CenterPoint (voxel) [6] ‡ | 57.00 | 45.32 | 31.66 | 27.14 | 40.47 | 37.23 | 20.14 | |
| Ours (pillar) ∗ | **62.97** | **53.31** | 34.62 | 26.56 | **31.61** | **26.02** | 18.71 | ✓ |
| Ours (pillar) ∗ | 62.80 | 53.20 | 34.62 | 26.56 | 31.62 | 26.07 | 19.10 | |
| Ours (voxel) ‡ | **66.10** | 58.73 | 33.31 | **26.32** | **28.80** | 25.11 | 19.08 | ✓ |
| Ours (voxel) ‡ | 66.04 | 58.62 | 33.33 | 26.34 | 28.80 | 25.11 | **19.06** | |

Table 2: Comparisons of different distillation approaches.

| Method | NDS ↑ | mAP ↑ | mATE ↓ | mASE ↓ | mAOE ↓ | mAVE ↓ | mAAE ↓ |
|---|---|---|---|---|---|---|---|
| Baseline (without distillation) | 62.80 | 53.20 | 34.62 | 26.56 | 31.62 | 26.02 | 19.10 |
| Feature distill (voxel→pillar) | 62.84 | 53.29 | 34.55 | 26.78 | 31.61 | 26.11 | 19.02 |
| Pseudo labels (voxel→pillar) | 63.18 | 53.31 | 34.58 | 26.55 | 31.29 | 25.99 | 18.87 |
| Set-to-set distill (voxel→pillar) | 63.37 | 53.89 | 34.34 | 26.25 | 31.01 | 25.57 | 18.77 |

# 6 Experiments

We present our experiments in four parts. We introduce the dataset, metrics, implementation, and optimization details in §6.1. Then, we demonstrate performance on the nuScenes dataset [66] in §6.2. We present knowledge distillation results in §6.3. Finally, we provide ablation studies in §6.4.

## 6.1 Training & testing procedures

**Dataset.** We experiment on the nuScenes dataset [66]. nuScenes provides rich annotations and diverse scenes. It has 1K short sequences captured in Boston and Singapore with 700, 150, 150 sequences for training, validation, and testing, respectively. Each sequence is ∼20s and contains 400 frames. This dataset provides annotation every 0.5s, leading to 28K, 6K, 6K annotated frames for training, validation, and testing. nuScenes uses 32-beam LiDAR, producing 30K points per frame. Following common practice, we use calibrated vehicle pose information to aggregate every 9 non-key frames to key frames, so each annotated frame has ∼300K points. The annotations include 23 classes with a long-tail distribution, of which 10 classes are included in the benchmark.

**Metrics.** The major metrics are mean average precision (mAP) and the nuScenes detection score (NDS). In addition, we use a set of true positive metrics (TP metrics), which include average translation error (ATE), average scale error (ASE), average orientation error (AOE), average velocity error (AVE), and average attribute error (AAE). These metrics are computed in the physical unit.

**Model architecture.** Our model consists of three parts: a point-based feature extractor, a DGCNN to encode object queries and to connect the point cloud features to object queries, and a detection head to output the categorical label and bounding box parameters. We experiment with PointPillars [4] and SparseConv [60] as feature extractors. The three blocks of the PointPillars backbone have $[3, 5, 5]$ convolutional layers, with dimensions $[64, 128, 256]$ and strides $[2, 2, 2]$; the input features are downsampled to 1/2, 1/4, 1/8 of the original feature map. For SparseConv, we use four blocks of $[3, 3, 3, 2]$ 3D sparse convolutional layers, with dimensions $[16, 32, 64, 128]$ and strides $[2, 2, 2, 1]$; the input features are downsampled to 1/2, 1/4, 1/8, 1/8 of the original feature map. For SparseConv, we transform the features into BEV by collapsing the $z$-axis. Both backbones use two deformable self-attention [17] layers with dimensions $[256, 256]$ to transform the BEV features. Then, we use two DGCNNs to encode the object queries. Each DGCNN [8] contains two EdgeConv layers with dimensions $[256, 256]$, both with 16 nearest neighbors. For each object query, we predict four points

Table 3: Comparisons of self-distillation versus baselines.

| Method | NDS ↑ | mAP ↑ | mATE ↓ | mASE ↓ | mAOE ↓ | mAVE ↓ | mAAE ↓ |
|---|---|---|---|---|---|---|---|
| Baseline (pillar, without distillation) | 62.80 | 53.20 | 34.62 | 26.56 | 31.62 | 26.02 | 19.10 |
| Self-distillation (pillar→pillar) | 63.41 | 53.89 | 34.21 | 26.19 | 31.11 | 25.67 | 18.54 |
| Baseline (voxel, without distillation) | 66.04 | 58.62 | 33.33 | 26.34 | 28.80 | 25.11 | 19.06 |
| Self-distillation (voxel→voxel) | 66.45 | 59.25 | 31.17 | 25.77 | 30.73 | 25.72 | 18.77 |

Table 4: Self-distillation with privileged information.

| Method | NDS ↑ | mAP ↑ | mATE ↓ | mASE ↓ | mAOE ↓ | mAVE ↓ | mAAE ↓ |
|---|---|---|---|---|---|---|---|
| Sparse→sparse (pillar) | 42.12 | 38.89 | 44.01 | 27.78 | 64.01 | 144.01 | 39.21 |
| Dense→sparse (pillar) | 42.79 | 39.10 | 43.89 | 27.77 | 64.01 | 143.97 | 39.11 |
| Sparse→sparse (voxel) | 59.55 | 49.84 | 31.17 | 25.77 | 33.73 | 32.22 | 20.22 |
| Dense→sparse (voxel) | 59.89 | 50.12 | 31.11 | 25.76 | 33.70 | 32.19 | 20.11 |

in the BEV to obtain and aggregate the BEV features. The final feature for this object query is the weighted sum of features of these four BEV points. The final detection head takes the features of each object query and predicts class label and bounding box parameters w.r.t. the reference point.

**Training & inference.** We use AdamW [67] to train the model. The weight decay for AdamW is $10^{-2}$. Following a cyclic schedule [68], the learning rate is initially $10^{-4}$ and gradually increased to $10^{-3}$, which is finally decreased to $10^{-8}$. The model is initialized with a pre-trained PointPillars network on the same dataset. We train for 20 epochs on 8 RTX 3090 GPUs. During inference, we take the top 100 objects with highest classification scores as the final predictions.We *do not* use any post-processing such as NMS. For evaluation, we use the toolkit provided with the nuScenes dataset.

## 6.2 Object DGCNN

We compare to top-performing methods on the nuScenes dataset in Table 1. PointPillars [4] is an anchor-based method with reasonable trade-off between performance and efficiency. FreeAnchor [64] extends PointPillars by learning how to assign anchors to the ground-truth. RegNetX-400MF-SECFPN [65] uses neural architecture search (NAS) to learn a flexible neural network for 3D detection; it is essentially a variant of PointPillars with an enhanced backbone network. Different from anchor-based methods, Pillar-OD [5] makes predictions per pillar, alleviating the class imbalance issue caused by anchors. CenterPoint [6] exploits similar detection heads, with better performance using better training scheduling and data augmentation. For these methods, we use re-implementations in MMDetection3D [69], which match the performances in the original papers.

We mainly compare to CenterPoint with both PointPillars and SparseConv backbones, denoted as "voxel" and "pillar" respectively. Our method outperforms other methods significantly including CenterPoint with NMS. Without NMS, the performance of CenterPoint drops considerably while our method is unaffected by NMS. This finding verifies the DGCNN implicitly models object relations and removes redundant boxes.

## 6.3 Set-to-set distillation

In this section, we present experiments involving our set-to-set distillation pipeline. We conduct three types of distillation. First, we distill a teacher model with a SparseConv backbone to a student model with a PointPillars backbone (denoted as "voxel→pillar"). This aligns with the common knowledge distillation setup for classification. We compare to feature-based distillation and pseudo label based methods. The objective of feature-based distillation is to align the middle-level features of the teacher model and the student model while the pseudo label based methods generate pseudo training examples with the pre-trained teacher networks. As Table 2 shows, our set-to-set distillation achieves better performance, confirming that distilling the last stage of the object detection model is more effective than distilling feature maps.

Second, we perform self-distillation [49] (denoted as "voxel→voxel" and "pillar→pillar"), where the teacher and the student are identical and take the same point clouds as input. As Table 3 shows, even when the teacher network and the student network have the same capacity, self-distillation still introduces a performance boost. This finding is consistent with the results in [49].

Finally, we try distillation with privileged information [56], where the teacher gets access to privileged information but the student does not. Following [10], the teacher takes dense point clouds, and the

student takes sparse point clouds (denoted as "dense→sparse"). To limit computation time, we train each model over a shorter period of time. The goal is for the student model to learn the same representations as the teacher model without knowing the dense inputs. In Table 4, we compare this setup with self-distillation, where the difference is the teacher model and the student model take the same sparse point clouds in self-distillation. The student achieves better performance when the teacher takes dense point clouds. The result suggests that set-to-set knowledge distillation is an effective approach to transfer insight from privileged information.

## 6.4 Ablation

We provide ablation studies on different components of our model to verify assorted design choices. First, we study the improvements of DGCNN over its counterpart, multi-head self-attention [9]. The multi-head self-attention has 8 heads with embedding dimension 256 and LayerNorm [70], following common usage. The DGCNN has two EdgeConv layers with dimensions $[256, 256]$. The number of neighbors $K$ in EdgeConv is 16. In principle, DGCNN is a sparse version of multi-head self-attention; the sparse structure reduces overhead in back-propagation and leads to sharper "attention maps" as well as faster convergence.

Table 5: DGCNN versus multi-head self-attention.

| Method / Metric | Multi-head self-attention | DGCNN |
|---|---|---|
| NDS | 39.89 | **41.32** |
| mAP | 36.35 | **37.81** |

Table 6: Models with different # DGCNNs.

| # layers / Metric | 1 | 2 | 3 | 4 | 5 | 6 |
|---|---|---|---|---|---|---|
| NDS | 35.91 | 39.75 | 41.15 | 41.26 | 41.07 | **41.32** |
| mAP | 32.32 | 36.54 | 37.25 | 37.75 | 37.78 | **37.81** |

Table 7: The number of neighbors in DGCNN.

| # neigbhors / Metric | 1 | 4 | 8 | 16 | 32 | 64 |
|---|---|---|---|---|---|---|
| NDS | 40.21 | 40.45 | 40.51 | **41.32** | 40.17 | 39.80 |
| mAP | 36.81 | 37.15 | 37.46 | **37.81** | 37.12 | 36.70 |

Table 8: The distribution of the output scores with respect to overlapping boxes.

| Method | filtered boxes | remaining boxes | all boxes |
|---|---|---|---|
| CenterPoint | 0.0764 | 0.1859 | 0.0829 |
| Ours | 0.1222 | 0.1711 | 0.1604 |

Table 9: Complexity comparsion between DGCNN and Multi-head self-attention.

| Module | Complexity | # parameters |
|---|---|---|
| DGCNN | $O(n^2 d)$ | 262144 |
| Multi-head self-attention | $O(n^2 d)$ | 263168 |

Table 5 shows the comparisons: DGCNN consistently outperforms multi-head self-attention. This aligns with our hypothesis: objects are distributed sparsely in the scene, so dense interactions among objects are neither efficient nor effective. Furthermore, we study the effect of number of neighbors in DGCNN. When it is 1, The model reduces to an architecture without object interaction. As we increase the number, it approaches multi-head self-attention. As shown in Table 7, the sweet spot is 16, which appears to balance object interactions and sparsity.

We also investigate improvements introduced when more DGCNNs are stacked in Table 6. This result suggests it is beneficial to incorporate multiple DGCNNs to model the dynamic object relations.

Moreover, we hypothesize our method produces different distribution of the output scores with respect to overlapping boxes. To verify this hypothesis (Table 8, we compute average scores for three types of boxes: filtered boxes after NMS, remaining boxes after NMS, and all boxes. Below we show the results. We also compute the percentage of filtered boxes by NMS in our method and Centerpoint. In our method, 21.9% boxes are removed while in Centerpoint 85.16% boxes are filtered. Hence, we conclude that our method indeed exhibits a different distribution pattern from Centerpoint.

Finally, we include a complexity comparison between DGCNN and Multi-head self-attention. Table 9 shows the results; DGCNN layer is on a par with Multi-head self-attention.

## 7 Conclusion

Object DGCNN is a highly-efficient 3D object detector for point clouds. It is able to learn object interactions via dynamic graphs and is optimized through a set-to-set loss, leading to NMS-free detection. The success of Object DGCNN indicates that many post-processing operations in 3D object detection are likely unnecessary and can be replaced with suitable neural network modules. Moreover, we introduce a set-to-set knowledge distillation pipeline enabled by the Object DGCNN. This new pipeline significantly simplifies knowledge distillation for 3D object detection and may

be applicable to other tasks like 3D model compression. Beyond the direct usage of our model, our experiments suggest several future directions to address current limitations. For example, our method is initialized with a pre-trained backbone network. Training the model from scratch remains elusive due to the sparse set-to-set supervision; solving this issue may yield improved generalization as in [71]. Furthermore, studying 3D-specific feature extractors will improve the speed and generalizability of 3D object detection. Finally, the large amount of unlabeled data available at training time can serve as another type of privileged information to apply self-supervised learning to 3D domains through set-to-set distillation.

**Potential impact.** Our method aims to improve the object detection pipeline, which is crucial for the safety of autonomous driving systems. One potential negative impact of our work is that it still lacks theoretical guarantees, similar to many deep learning methods. Future work to improve applicability in this domain might consider challenges of *explainability* and *transparency*.

## 8 Acknowledgement

The MIT Geometric Data Processing group acknowledges the generous support of Army Research Office grants W911NF2010168 and W911NF2110293, of Air Force Office of Scientific Research award FA9550-19-1-031, of National Science Foundation grants IIS-1838071 and CHS-1955697, from the CSAIL Systems that Learn program, from the MIT–IBM Watson AI Laboratory, from the Toyota–CSAIL Joint Research Center, from a gift from Adobe Systems, from an MIT.nano Immersion Lab/NCSOFT Gaming Program seed grant, and from the Skoltech–MIT Next Generation Program.

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
