# OpenReview forum: "Object DGCNN: 3D Object Detection using Dynamic Graphs"
_NeurIPS.cc/2021/Conference — NeurIPS 2021 Poster_

### Official Review · Reviewer_UvFF · 2021-07-14

**Rating:** 6
**Confidence:** 3

**Summary:**

The paper introduces an NMS-free framework for 3D object detection in self-driving scenarios, following the 2D representative work DETR. Instead of using Transformers, it uses a dynamic graph CNN (DGCNN) to generate the set of detected objects. Besides, it proposes a set-to-set distillation loss to distill the knowledge of the teacher model, e.g., trained on multiple frames, for the student model, which will be tested on single frames. The experiments are only conducted on nuScenes.

**Limitations And Societal Impact:**

The authors have not adequately addressed the limitations.

**Main Review:**

Strongness:
- The writing is clear.

---

Weakness:
1. The major contributions are ambiguous. The connection between the NMS-free detector and the distillation is not tight. It confuses me when I read the introduction for the first time, as the knowledge distillation seems to suddenly appear. The authors seem to plan to equip the paper with two contributions, but it turns out that they are not connected well. Motivations, especially the connection of the two contributions, can be discussed more in the introduction.
2. The experiments are only done on one dataset (nuScenes). Most 3D object detection papers recently conduct experiments on at least two datasets (KITTI, Waymo apart from nuScenes), or conduct more comprehensive experiments on one dataset, in order to make the results more convincing. Can the authors show the results on any other dataset?
3. For the distillation experiments, the models are trained only for 1 epoch (mentioned in Sec 6.3). Can the authors report the results in Sec 6.3 and 6.4 with longer training time (like 20 epochs), so that they can be directly compared with the numbers in Table 1?
4. As the authors have mentioned in Sec 6.4, dense graph networks, like Transformers, might be slower to converge compared to sparse graph networks, like DGCNN. Thus, can the authors also report the ablation results with a longer training time (20 epochs)?


---

Typo:
L55: objectors -> object detectors
The introduction section title is missing.

---

**Updates after rebuttal**
I slightly raise my rating to 6 according to the authors' rebuttal. However, the authors should improve their connection between two contributions.



**Time Spent Reviewing:**

2

---

> ### Author Response · Authors · 2021-08-10
> **Thank you for the thoughtful questions! We address the questions below.**
>
> Thank you for taking the time to consider our paper and asking thoughtful questions to help improve our paper. We addresses the questions below.
>
> Q1. Contribution
>
> The major contribution of this paper is the introduction of an NMS-free 3D object detector for point clouds. We equip the model with DGCNN and set-to-set prediction, which makes it possible to alleviate the need for NMS while achieving comparable (or better) performance to SOTA. In addition, our model enables a simple approach to distillation via set-to-set distillation. This distillation is not trivial using previous architectures such as anchor-based prediction or pillar-based prediction.
>
> Q2. Additional experiments on KITTI
> We provide additional results on KITTI below. We compare our method to PointPillars and SECOND, which are faithfully reimplemented in MMDetection3D. We use the same backbone with SECOND for a fair comparison. Our method achieves comparable performance to SECOND even though we do not perform extensive hyperparameter tuning. We report 3D mAP with easy/moderate/hard breakdowns. We will add results on Waymo in the revision if time permits.
>
> | Method | easy | moderate | hard |
> |-----------|---------|--------------|--------|
> | PointPillars | 71.58 | 59.50 | 55.02 |
> | SECOND | 75.21 | 64.41 | 60.73 |
> | Ours | 74.29 | 65.17 | 60.55 |
>
> Q3. Longer training of distillation experiments
>
> In the following table, we provide the distillation results with longer training (20 epochs). We compare the model without distillation, a pillar-to-pillar distilled model, a voxel-to-pillar distilled model. Both models with distillation outperform the one without distillation.
>
> | Method | mAP | NDS |
> |-----------|---------|--------|
> | Baseline | 48.71 | 60.54 |
> | Pillar-to-pillar | 49.88 | 60.75 |
> | Voxel-to-pillar | 49.33 | 60.61|
>
> Q4. Longer training of DGCNN vs. multi-head self-attention.
>
> We provide a comparison with a longer training schedule (20 epochs). When training longer, the one with DGCNN significantly outperforms its counterpart.
>
> | Model | mAP | NDS |
> |----------|--------|--------|
> | DGCNN | 48.71 | 60.54|
> | Multi-head self-attention | 43.44 | 56.08 |
>
> Please let us if our responses are sufficient. We are happy to address any remaining points in the discussion. We kindly ask you to consider our response and raise your score if all questions are addressed.

---

> > ### Comment · Reviewer_UvFF · 2021-08-16
> > **Update Table 2 and 3**
> >
> > > Q3. Longer training of distillation experiments
> > Can the authors update the whole Table 2 and Table 3 results with longer training epochs here? It is a little difficult to understand the current table in the rebuttal.

---

> > > ### Author Response · Authors · 2021-08-17
> > > **Updated Table 2 and Table 3**
> > >
> > > Thank you for the follow-up questions. Below we update the corresponding tables with new results. Please let us know if you have further questions; we are happy to provide additional results or discussion to address any concerns.
> > >
> > > Table 2
> > >
> > > | Method | mAP | NDS |
> > > |-----------|---------|--------|
> > > | Baseline (pillar, without distillation) | 48.71 | 60.54 |
> > > | Feature distill (voxel-to-pillar) | 48.81 | 60.37 |
> > > | Pseudo labels (voxel-to-pillar)| 47.40 | 59.55 |
> > > | Set-to-set distill (voxel-to-pillar) | 49.33 | 60.61|
> > >
> > > Table 3
> > >
> > > | Method | mAP | NDS |
> > > |-----------|---------|--------|
> > > | Baseline (pillar, without distillation)| 48.71 | 60.54 |
> > > | Self-distillation (pillar, pillar-to-pillar) | 49.88 | 60.75 |
> > > | Baseline (voxel, without distillation) | 54.87 | 64.18 |
> > > | Self-distillation (voxel, voxel-to-voxel) | 55.21 | 64.47 |

---

### Official Review · Reviewer_j8MH · 2021-07-16

**Rating:** 6
**Confidence:** 4

**Summary:**

This paper proposes a method for object detection and velocity estimation in 3D LIDAR scans.  It improves previous voxel- or pillar-based frameworks by adding a graph network and by avoiding non-maximum suppression by teaching the network to not produce duplicates through bipartite result-GT-matching. Additionally, the bipartite matching is used to design a knowledge destillation framework to create a further improved network.

The proposed design is evaluated on a large standard dataset where it reaches and sometimes surpasses the performance of state of the art, without requireing an explicit NMS. Additional ablation studies show the effectiveness of the individual building blocks.

**Ethical Concerns:**

There are no ethical issues with this paper.

**Limitations And Societal Impact:**

Limitations, potential further work and potential societal impact are discussed in the conclusion.


**Main Review:**


## Overview

The paper introduces a well though off architecture that introduces a NMS-free way for 3D object detection in LIDAR data. Though somewhat incremental in nature, the method is well motivated and convincingly evaluated with strong results. The end-to-end trainable architecture might open further research, as demonstrated with the proposed knowledge destillation.

## Details

* Overall the paper is very well written and and method is well described and can likely be reproduced based on the manuscript.

  The only part I cannot quite follow are the formalities in Eq.(9) and (10). Shouldn't one of the 1_{...} expressions be c_j != empty set, instead of c_j == empty set? What is N in Eq.(10),(11),(12) (in \Sum), is it the same N as line 131? Is the M in Eq.(9) the original number of ground-truth boxes, or the "extended" number M^*? To be honest it is rather difficult for me to follow these equations and their motivation and impact without looking into [7].

* Though none of the building blocks used in the paper is novel ([4] and [60] as backbone, DGCNN for graph processing, [7] for the NMS-free bipartite matching, and a common student-teacher knowledge destillation), the combination is non-trivial and well thought off. The knowledge destillation would not be possible that easily without the NMS-free architecture, and the DGCNN built upon the Bird-Eye-View representation of the LIDAR data is well though off, aiding computational costs without much loss of performance, combining spatial relationships with feature-based relations through the dynamic graph. The method is quite generic and can use multiple different backbones. The NMS-free architecture might open up additional research.

* The experiments are fair, extensive and overall convincing. A large standard dataset with a common evaluation protocoll is used, and the method is compared against state of the art. Several ablation studies are peformed to evaluate the impact of the different proposed extensions.

* The results are good. In terms of absolute performance, the method is en par with or slightly exceeds state of the art. The results demonstrate that the proposed NMS-free architecture is a good replacement for NMS, i.e., the network learns not to produce duplicate results.

  The only experiment that might be missing is to separately evaluate the impact of the Object DGCNN and the bipartite matching. For this, the last block of the network (the bipartite matching) could be replaced by a soft assignment followed by NMS, to compare "Object DGCNN + NMS" vs. "Object DGCNN + Bipartite, no NMS" (from what I understand, the methods that include NMS Table 1 evaluate "Object DGCNN + Bipartite + NMS").

* An additional potentially interesting experiment is the distribution of the output scores w.r.t. overlapping boxes. The intuition is that the network learns the NMS and produces a single high-confidence prediction per object. Since a fixed number of predictions is used during inference, it is possible and even likely that these might include duplicate predictions, though with a lower confidence. If this intuition is correct, it could be demonstrated for example by performing a NMS and showing the confidence distributions of the removed boxes compared to the kept box; this should show a different curve than when performing NMS on [6], for example.


## Further Notes

Typos etc., no need to respond.

* Eq.(3) and l.183: \delta^0 ... \delta^K is K+1 points, not K (compare L in l.173)

* l.132: A more precise list of the 9 features of the bounding box would be interesting, or a reference to some paper that describes it.

* The following paper might be interesting in the related work section. It is from this year's CVPR, i.e. published after the submission deadline, so it's not a must and completely at the authors' discretion:

    [R1] Kumar, Abhinav, Garrick Brazil, and Xiaoming Liu. "GrooMeD-NMS: Grouped Mathematically Differentiable NMS for Monocular 3D Object Detection." Proceedings of the IEEE/CVF Conference on Computer Vision and Pattern Recognition. 2021.


**Time Spent Reviewing:**

5

---

> ### Author Response · Authors · 2021-08-10
> **Thank you for providing the detailed review and the constructive feedback!**
>
> Thank you so much for the constructive feedback and thoughtful questions! We are really glad you like our paper. We provide responses to specific questions.
>
> Q1. Clarification of Eq (9), (10), (11), and (12).
>
> It should be c_j != empty set in Eq (9) and (10). M* is the number of predictions while M is the number of ground-truth boxes. In Eq (11) and (12), N is the number of predictions (which we can write as M*). We will clean up the notation in the revision. Thank you for pointing out these typos.
>
> Q2. Compare "Object DGCNN + NMS" vs. "Object DGCNN + Bipartite, no NMS"
>
> We design an additional experiment on the "Object DGCNN + NMS" setting. First, we remove the bipartite matching in our pipeline. Then, we use softmax to normalize the distance (a combination of classification loss and regression loss as in our method) between predictions and the ground-truth boxes. Finally, we soft-assign a prediction to the ground-truth boxes in training. In inference, we perform NMS. We observe unstable training in this setting and lower the learning rate by 10. The following table shows the comparison. The one without bipartite matching significantly underperforms our method.
>
> | Method | mAP | NDS |
> |-----------|---------|--------|
> |Object DGCNN + Bipartite, no NMS| 48.71 | 60.54|
> |Object DGCNN + NMS|42.56 | 54.65|
>
> Q3. The distribution of the output scores w.r.t. overlapping boxes
>
> We compute average scores for three types of boxes: filtered boxes after NMS, remaining boxes after NMS, and all boxes. Below we show the results. We also compute the percentage of filtered boxes by NMS in our method and Centerpoint. In our method, 21.9% boxes are removed while in Centerpoint 85.16% boxes are filtered. Hence, we conclude that our method indeed exhibits a different distribution pattern from Centerpoint.
>
> |Method |  filtered boxes | remaining boxes | all boxes |
> |----------|---------------------|-----------------------|---------------|
> | CenterPoint | 0.0764 | 0.1859 | 0.0829 |
> | Ours | 0.1222 | 0.1711 | 0.1604 |

---

> > ### Comment · Reviewer_j8MH · 2021-08-27
> > **Follow-Up Question**
> >
> > Thank you for the additional experiments. Could you please clarify the methods in the Table of Rebuttal, Q2 and Table 1 of the submitted manuscript. The values in Q2 for "Object DGCNN + NMS" match the values in Table 1, "Ours (pillar)", I was under the impression that the results in Table 1 already include Bipartite matching. In other words, I thought the last 4 rows in Table 1 were
> >
> > * Pillars -> Object DGCNN + Bipartite + NMS
> > * Pillars -> Object DGCNN + Bipartite + no NMS
> > * Voxel -> Object DGCNN + Bipartite + NMS
> > * Voxel -> Object DGCNN + Bipartite + no NMS
> >
> > but apparently they are not.

---

> > > ### Author Response · Authors · 2021-08-27
> > > **Typo**
> > >
> > > Thank you so much for point out this! It was a typo in the previous response. We apologize for that. Your understanding about the last 4 rows in Table 1 is absolutely correct. The table in the previous response (fixed) should be:
> > >
> > > | Method | mAP | NDS |
> > > |-----------|---------|--------|
> > > |Object DGCNN + Bipartite, no NMS| 48.71 | 60.54|
> > > |Object DGCNN + NMS|42.56 | 54.65|
> > >
> > > So without bipartite matching, the performance drops a lot even the model is equipped with NMS.

---

### Official Review · Reviewer_vpe9 · 2021-07-17

**Rating:** 6
**Confidence:** 4

**Summary:**

This paper focuses on the task of 3D object detection from point cloud. Inspired by recent non-maximum suppression-free 2D object detection modes, this paper proposes a new 3D object detection architecture. Concretely, it models 3D object detection as message passing on a dynamic graph, generalizing the framework to predict a set of objects. Furthermore, it removes the necessity of post-processing via object confidence aggregation or non-maximum suppression. To facilitate object detection from sparse point clouds, the authors also propose a set-to-set distillation approach customized to 3D detection. Experiments are conducted on nuScenes dataset to demonstrate the effectiveness of the proposed method.

**Limitations And Societal Impact:**

The author has described the limitations of the work in Sec. 7. Please see comments above for more details

**Main Review:**

1、The contribution of this paper is limited, and the motivation from the first section seems to be not solid.

First, the authors just very simply introduce methods of PointPillars, PillarOD and CenterPoint in 3D object detection(L19-26), which is obviously not enough. Directions like voxel/point-based or one/two-stage are not discussed.

Second, the proposed method mainly involves technologies of set-to-set prediction [9], GCN [8] and knowledge distillation [10], all of them are popular algorithms published by others recently. However, the author didn’t not provide detailed and reasonable discussions/comparisons about why using these technologies, it seems to just simply combine them together. For the part of knowledge distillation, what is the difference or improvement comparing with [10, 56, 49]? What is the insightful design in sparse 3D object detection comparing with DETR?

For references, some sparse object detection methods [a,b] and graph-based 3d detection methods [c,d] are missing in the section of Related Work

2、In Tab. 5, to compare the performance between DGCNN and multi-head self-attention, it would be better to list the computational cost, like FLOPs and #Parameters. Is the depth of multi-head self-attention the same as DGCNN (i.e., 6)?

3、The ablation study is insufficient. In Sec. 6.4, most of them are about parameter tuning. It lacks the performance analysis of different modules (e.g., set-to-set prediction, DGCNN and knowledge distillation). For example, what are the advantages of the structural design of the proposed modules comparing with variations or similar competitors? Are there any insightful conclusions to be drawn from the optimal structural design or is the design consistent with the motivation or original idea?


[a] Sun, Peize, et al. "Sparse r-cnn: End-to-end object detection with learnable proposals." CVPR2021

[b] Liu, Zili, et al. "SparsePoint: Fully End-to-End Sparse 3D Object Detector."

[c] Chen, Jintai, et al. "A hierarchical graph network for 3D object detection on point clouds." CVPR2020

[d] Shi, Weijing, and Raj Rajkumar. "Point-gnn: Graph neural network for 3d object detection in a point cloud." CVPR2020

After reading the authors' rebuttal and other reviewers' comments, they have solved part of my concerns, thus I would like to raise the rating to 6.

**Time Spent Reviewing:**

1.5

---

> ### Author Response · Authors · 2021-08-10
> **Thank you for the thoughtful questions! We address the specific questions raised in the review.**
>
> Thank you for the detailed review and questions. Below we address the specific questions.
>
> Q1. Motivation, novelty, comparisons to [10, 56, 49], and related works [a, b, c, d].
>
> The high-level goal of our paper is to design a post-processing free 3D object detector. To that end, we made several design choices: 1. We use DGCNN to model object-object interactions so objects have information about their peers. 2. The set-to-set loss enforces the model learns to reduce redundancy. As a byproduct, knowledge distillation becomes extremely straightforward to implement with the set-to-set sparse structure. To the best of our knowledge, no paper has used DGCNN or related architectures to model object relationships and our paper is the first attempt of its kind to make post-processing 3D perception possible. Also as reviewer j8MH mentions, the combination of DGCNN, PonitPillars, and set-to-set distillation is non-trivial and well-suited to our target application.
>
> Our method processes highly sparse point clouds compared to DETR, which deals with 2D images. Objects are less sparse in 2D images than their counterparts in 3D point clouds, which motivates the design of a new sparse relation module (DGCNN).
>
> [10], which is a specific example of [56], uses a feature-based approach to distill information between a teacher model and a student model. We compare to a similar baseline (“Feature distill” in Table 2) and show that our set-to-set distillation is more effective. [49] focuses on self-distillation and demonstrates that a sequence of self-distillation steps can be beneficial. We perform self-distillation (in Table 2, Table 3, and Table 4), which is motivated by [10, 56, 49], to demonstrate that our paper enables a simple set-to-set distillation pipeline.
>
> We will add [a, b, c, d] in the revision.
>
> Q2. "it would be better to list the computational cost, like FLOPs and #Parameters. Is the depth of multi-head self-attention the same as DGCNN (i.e., 6)?"
>
> The depth of multi-head self-attention is indeed the same as DGCNN. We use standard multi-head self-attention with 256 hidden dimensions and 8 heads. In the DGCNN, we use 2 EdgeConv layers with 256 hidden dimensions. We list # parameters of one DGCNN layer and one multi-head self-attention in the following table. Due to time constraints, we cannot measure the exact FLOPs, but the complexity of DGCNN (KNN graph construction) is the same as that of multi-head self-attention (dot product). (n is the number of objects and d is the number of hidden dimensions)
>
> | Module | Depth | # parameters |
> |-----------|----------|-----------|
> | DGCNN | O(n^2 d) | 262144|
> | Multi-head self-attention | O(n^2 d) | 263168|
>
> Q3. "What are the advantages of the structural design of the proposed modules comparing with variations or similar competitors? Are there any insightful conclusions to be drawn from the optimal structural design or is the design consistent with the motivation or original idea?"
>
> As mentioned above, objects are sparsely distributed in a point cloud scene and are distant from each other. DGCNN allows us to model such sparse relations, in contrast to multi-head attention, where objects are densely connected. In our experiments (Table 7), we demonstrate that DGCNN indeed outperforms multi-head attention. The conclusion we draw is that a 3D object detector with sparse structures incorporated throughout is superior and has lower computation overhead in inference.
>
> Please let us know if our responses addresses your concerns. We are more than happy to address to any further questions during the discussion period. If the sufficient responses are sufficient, we kindly ask that you consider raising your score.

---

> > ### Author Response · Authors · 2021-08-21
> > **Follow-up**
> >
> > Thank you again for your constructive comments and suggestions. If we have successfully addressed your questions, we would strongly appreciate an increased score. Otherwise, please let us know and we are happy to provide additional experiments and/or discussion to allay your concerns.

---

### Decision · Program_Chairs · 2021-09-27

**Decision:**

Accept (Poster)

**Comment:**

After rebuttal, all three expert reviewers came to the same conclusion, namely that the paper would be a good contribution to NeurIPS. Having examined the reviews, rebuttal, and post-rebuttal discussion, the AC is in agreement with the reviewers. Given that the additional experiments (e.g., ablations) and promises of clarifications from the rebuttal were important for accepting the paper, the AC highly encourages the authors to incorporate these additional results into the final version of the paper.